# A beta-Poisson model for infectious disease transmission

**Joe Hilton**[1], **Ian Hall**[2]*

**1** School of Life Sciences and Zeeman Institute (SBIDER), University of Warwick, Coventry, United Kingdom, **2** Department of Mathematics and School of Health Sciences, University of Manchester, Manchester, United Kingdom

\* ian.hall@manchester.ac.uk

**Data Availability Statement:** All code used in this study is available at https://github.com/JBHilton/beta-poisson-epidemics/ under an MIT license.

**Funding:** The author(s) received no specific funding for this work.

## Abstract

Outbreaks of emerging and zoonotic infections represent a substantial threat to human health and well-being. These outbreaks tend to be characterised by highly stochastic transmission dynamics with intense variation in transmission potential between cases. The negative binomial distribution is commonly used as a model for transmission in the early stages of an epidemic as it has a natural interpretation as the convolution of a Poisson contact process and a gamma-distributed infectivity. In this study we expand upon the negative binomial model by introducing a beta-Poisson mixture model in which infectious individuals make contacts at the points of a Poisson process and then transmit infection along these contacts with a beta-distributed probability. We show that the negative binomial distribution is a limit case of this model, as is the zero-inflated Poisson distribution obtained by combining a Poisson-distributed contact process with an additional failure probability. We assess the beta-Poisson model's applicability by fitting it to secondary case distributions (the distribution of the number of subsequent cases generated by a single case) estimated from outbreaks covering a range of pathogens and geographical settings. We find that while the beta-Poisson mixture can achieve a closer to fit to data than the negative binomial distribution, it is consistently outperformed by the negative binomial in terms of Akaike Information Criterion, making it a suboptimal choice on parsimonious grounds. The beta-Poisson performs similarly to the negative binomial model in its ability to capture features of the secondary case distribution such as overdispersion, prevalence of superspreaders, and the probability of a case generating zero subsequent cases. Despite this possible shortcoming, the beta-Poisson distribution may still be of interest in the context of intervention modelling since its structure allows for the simulation of measures which change contact structures while leaving individual-level infectivity unchanged, and vice-versa.

## Author summary

The early stages of epidemics are characterised by dramatic variations in the number of new cases generated by each infectious individual, with some cases generating no new infections and some "superspreading" cases generating disproportionately large numbers

**Competing interests:** The authors have declared that no competing interests exist.

of subsequent cases. In this study we introduce a mathematical model based on a two-step interpretation of infectious disease transmission: infectious individuals make a random number of contacts according to some fixed contact distribution and then infect their contacts with an infection probability which is unique to that specific infectious individual. This model has the advantage of generalizing more commonly used models of early epidemic dynamics, while allowing for policy analyses which assess the impact of measures which impact social contact behaviour and infectiousness across contacts separately. We find that while our model performs at least as well as pre-existing models in modelling individual-level capacity to generate new infections, the extra mathematical complexity our model introduces is not justified by commonly-used measures of parsimony. This suggests that our model could be applicable in specific policy settings but does not offer a substantial improvement over past approaches in a purely observational setting.

## Introduction

Infections at the human-animal interface are often associated with high levels of morbidity and mortality, making them a subject of substantial interest to mathematical modellers and other infectious disease researchers [1]. Outbreaks of these infections are often characterised by a combination of low reproductive ratios, with most cases infecting only a small number of contacts, and the presence of *superspreading* events in which a few individual cases generate disproportionately large numbers of subsequent infections [1–3]. Similar dynamics are also characteristic of outbreaks of infections such as measles and mumps in populations which are mostly vaccinated and where the infection is close to eradication [1, 2]. While low rates of overall spread tend to make outbreaks of novel pathogens relatively small and short-lived in comparison to outbreaks of more well-established seasonal or endemic pathogens, the potential for superspreading poses a challenge in predicting the dynamics of ongoing outbreaks. Superspreading events pose a unique public health challenge in that failure to effectively intervene in an outbreak can result in rapid and unexpected increases in case numbers. An important tool in the control of small localised outbreaks is contact tracing, through which public health professionals aim to identify social contacts of already-identified cases, allowing for rapid diagnosis of potential cases along with treatment and, if necessary, quarantine to minimise further infection [4, 5]. Contact tracing also offers a vital source of scientific information by identifying specific infector-infectee pairs from which we can infer patterns of individual-level variation in transmission [6]. Where multiple generations of infector-infectee pairs can be identified, the observed cases can be assigned a position in a *transmission chain*, a network tracing each case back to its infector. The topology of this network characterises the epidemic: the mean number of onward connections from each case gives the basic (in the case of a novel infection with no preexisting immunity) or effective (in the close-to-eradication setting) reproductive ratio of the outbreak [2].

The key to interpreting transmission chain behaviour from a mathematical perspective is the theory of *branching processes*. Branching process models provide a rigorous description of an outbreak's early behaviour in the stages before the underlying susceptible population becomes depleted [7] and are a well-established item in the inventory of mathematical methods available to epidemiologists [1, 3, 8–13]. A branching process model can be formulated from transmission chain data by using the degree distribution of the transmission network as the branching process's *offspring distribution*, the distribution of the number of cases generated by each case. We refer to these cases as *secondary cases*. In some cases rather than a fully

described transmission chain we may just know the number of secondary cases traced back to each case in the outbreak, and we will refer to such data as *secondary case data*. Results from the theory of branching processes can be used to calculate useful quantities like the probability that an outbreak goes extinct and the probability that it reaches a given size before extinction [14]. One of the major benefits of branching process models over more typical compartmental models is that they are not tied to the combination of a Poisson contact process and exponential infectious period which underlies most compartmental modelling. Although models for infections with non-exponential infectious periods can be simulated and analysed, they are in general non-Markovian and call for sophisticated mathematical techniques [15]. In particular, branching processes whose offspring distribution is the same at every generation, known as Galton-Watson processes, are always Markovian since the number of cases in a given generation depends only on the number in the last generation and are easily simulated on this generation-by-generation basis by drawing successive random numbers from the offspring distribution.

While some studies use the empirical transmission chain degree distribution as an offspring distribution [11, 16], it is more common to use a parametric model. Commonly used offspring distributions include the Poisson [8], geometric [17], and negative binomial [2, 3, 12, 18] distributions. The geometric and negative binomial models are both examples of *mixed Poisson distributions* [19]. These are Poisson distributions where the Poisson parameter is allowed to vary according to some mixing distribution—in the case of the geometric and negative binomial, an exponential and a gamma distribution respectively. A geometric offspring distribution specifically captures the early behaviour of the homogeneous stochastic SIR model, where the exponentially-distributed infectious period and Poisson process contact behaviour combine to give an exponential-Poisson mixture, while the gamma mixing distribution in the negative binomial can similarly be understood in terms of the infectious period of an infection with multiple infectious stages [20]. Mixed Poisson distributions are always *overdispersed*, meaning their variance is larger than their mean, in contrast to the (unmixed) Poisson, whose mean and variance are equal [19]. The negative binomial model's overdispersion gives it the capacity to model superspreading events, where the number of secondary cases generated by a single case is substantially more than the mean. Lloyd-Smith *et al.* developed a more formal definition, where a superspreading event for an infection with effective reproductive ratio $R$ is a transmission event in the upper $n$th percentile (in their study they take $n = 99$) of the Poisson distribution with mean $R$ [2]. Examples of superspreading have been recorded in novel coronavirus outbreaks [21, 22], and in the close-to-eradication setting in measles [23].

An alternative approach to modelling overdispersed count data is to use *zero inflation*. Zero inflated distributions specifically model count data with more zeros than would be expected under ordinary modelling assumptions [24, 25]. This is a natural interpretation to consider since transmission chains often contain a high proportion of individuals who do not produce any subsequent cases. In this context zero inflation can be interpreted as modelling situations in which infectious cases are unable to engage in ordinary contact behaviour either because of hospitalization and effective control measures, or simply because they are too unwell.

Mechanistic models of infectious disease dynamics are useful not just in their ability to summarise observed patterns of infection, but also in their ability to project the likely impact of public health measures designed to control infection. Interventions used during the COVID-19 pandemic highlighted the distinction between measures which act to reduce social contact rates, such as school and workplace closures, and those which act to reduce the likelihood of infection across a social contact, such as mask-wearing and vaccination. With this distinction in mind, in this study we introduce a beta-Poisson mixture model which offers a more mechanistically interpretable alternative to the negative binomial model. Whereas the negative

binomial encodes each infectious individual's propensity to generate secondary infections as a single gamma-distributed parameter, the beta-Poisson explicitly models a two-step process of contact and infection with individual-level variation in infectiousness, thus allowing for explicit modelling of control measures at both levels. Early formulations of the beta-Poisson mixture distribution date back to the 1960's [26], and in infectious disease modelling contexts it is sometimes used as a dose-response model [27, 28]. However, it is not typically used as a person-to-person transmission model despite its intuitive mechanistic interpretation and potential for modelling control measures. Our proposed beta-Poisson mixture approach is similar to a Poisson-generalized gamma mixture distribution which has been proposed as a more general alternative to the negative binomial in the context of COVID-19 modelling, but differs in that the generalized gamma mixing distribution lacks the beta distribution's interpretation as modelling individual-level variation in infectiousness across social contacts [29].

The aim of this study is to assess the beta-Poisson model's potential as a generic model of infectious disease transmission in comparison to the more commonly used but less mechanistically interpretable negative binomial model. To this end, we fit the beta-Poisson model to eight sets of secondary case data corresponding to a variety of pathogens and socio-economic settings. We compare the fitted beta-Poisson models with Poisson, geometric, negative binomial, and zero-inflated Poisson (ZIP) distributions fit to the same data and compare goodness-of-fit in terms of likelihood, Akaike Information Criterion (AIC), and more domain-specific metrics covering the models' abilities to capture the level of overdispersion, superspreading, and lack of onwards transmission seen in the data. In Section 1 of S1 Appendix we provide a full derivation of the beta-Poisson model's mathematical formulation, and demonstrate that both the negative binomial and ZIP models can be interpreted as special cases.

## Methods

### Model description

The beta-Poisson model simulates an epidemic starting from a set of index cases by assigning each case a Poisson-distributed number of contacts $y$ made during their infectious period and a beta-distributed transmission probability $p$ intended to capture individual-level differences in contact behaviour (for instance, the high frequency of physical touch-based contacts made by children relative to adults [30]) and physiological response (such as age-dependent effects in COVID-19 symptomaticity and individual-level variability in infectiousness in Ebola outbreaks [18, 31]). There are three model parameters: the mean number of secondary cases $\lambda$, a measure of overdispersion $\Phi$, and the mean number of contacts $N$ made over an individual's infectious period. Each case draws an individual-level transmission probability $p$ from a beta distribution with parameters $\alpha_1 = \lambda\Phi$ and $\alpha_2 = (N - \lambda)\Phi$, and infects each of their Poisson distributed number of contacts independently with probability $p$. The resulting number of secondary infections is thus a random variable $X$ such that $X \sim \mathrm{Bin}(Y, p)$ where $Y \sim \mathrm{Poi}(N)$ and $p \sim \mathrm{Beta}(\alpha_1, \alpha_2)$. If we condition on $p$ then this is precisely equivalent to a Poisson distribution with mean $pN$, i.e. $(X|p) \sim \mathrm{Poi}(pN)$, and so the statistics of the beta-Poisson model can be understood in terms of a Poisson distribution with a beta distributed scaling on the Poisson parameter without needing to explicitly consider the binomial draw step [32, Chapter 4]. Our formulation is a generalization of more standard definitions of the beta-Poisson distribution [26–28], where the Poisson parameter itself is $p$, equivalent to the specific case $N = 1$ in our model.

In Section 1 of S1 Appendix we demonstrate that four other common choices of secondary case distribution (Poisson, geometric, negative binomial, and zero-inflated Poisson) [2] can be considered special limiting cases of the beta-Poisson. In the limit $N \to \infty$, the beta-Poisson

distribution is equivalent to the negative binomial distribution with mean $\lambda$ and overdispersion $\theta = \Phi^{-1}$. Because the negative binomial distribution has the Poisson and geometric as special cases (overdispersion $\theta = 0$ and $\theta = \lambda$ respectively) these special cases are inherited by the beta-Poisson. The zero-inflated Poisson distribution is a convex combination of a Poisson distribution with mean $\tilde{\lambda}$ and a point probability mass at zero with convexity parameter $\sigma$, so that the probability of obtaining zero is given by $\sigma$ plus the probability of drawing zero under the Poisson distribution, and the probability of obtaining $x > 0$ is given by the Poisson probability of obtaining $x$ multiplied by $(1 - \sigma)$. The natural epidemiological interpretation of this distribution is that individuals either generate a Poisson number of secondary cases with probability $1 - \sigma$, or are prevented from making contacts due to either illness or a planned intervention and do not generate secondary cases with probability $\sigma$. This is analogous to our beta-Poisson distribution in the limit $\Phi \to 0$ (i.e. $\alpha_1 + \alpha_2 \to 0$). In this limit the underlying beta distribution tends to a two-point distribution with probability $\alpha_2/(\alpha_1 + \alpha_2)$ at zero and probability $\alpha_1/(\alpha_1 + \alpha_2)$ at one, so that the beta draw tends towards the choice in the ZIP distribution between generating zero cases or generating a Poisson-distributed number of case. The ZIP limit of the beta-Poisson distribution will have Poisson parameter $\tilde{\lambda} = N$ and zero inflation parameter $\sigma = 1 - \lambda/N$. A visual comparison of the mechanisms underlying the different offspring distribution models is provided in Fig 1. Definitions of all model parameters are provided for reference in Table 1.

The discrete contact and infection steps modelled by the beta-Poisson model offer an interpretive advantage over the negative binomial model, with the explicit contact parameter $N$ allowing for simulations of reductions in contact rates which preserve individual-level variation in contagiousness. In particular, in Section 5 of S1 Appendix we demonstrate that for an outbreak with beta-Poisson parameters $(\lambda, \Phi, N)$, an intervention which removes a proportion $(1 - N/\lambda)$ of all social contacts will be sufficient to end the outbreak. Despite this conceptual advantage, it is not obvious whether our model represents a substantial improvement over the simpler models outlined above in its accuracy as a descriptive model of transmission. To assess the beta-Poisson's ability to capture the patterns seen in data from real-world outbreaks in comparison to its limiting distributions, we will fit Poisson, geometric, negative binomial, ZIP, and beta-Poisson distributions to transmission chain data and compare their goodness of fit in terms of AIC. For each fitted distribution we additionally calculate the overdispersion, the probability that an individual does not generate any secondary cases, and the degree of superspreading, which we define to be the probability of generating a number of offspring in the upper 99th percentile of the fitted Poisson distribution following the definition used by Lloyd-Smith *et al.* [2].

For the Poisson and geometric models, the MLE of the single parameter $\lambda$ is given by the sample mean and so can be calculated directly. For the negative binomial model the MLE of $\lambda$ is given by the sample mean. We calculate the MLE of $\theta$ by finding the maximum of its log likelihood function numerically using the BFGS algorithm as implemented in the minimize function from the scipy.optimize package in Python [33]. For the zero-inflated Poisson model, the MLEs of $\lambda$ and $\sigma$ are calculated numerically using the BFGS algorithm; in this case $\lambda$ is not the mean of the overall distribution but the mean of the Poisson component of the distribution. For the beta-Poisson distribution, $\hat{\lambda}$ is given by the sample mean. In Section 1 of S1 Appendix we demonstrate that the contact parameter $N$ has to be at least as large as $\lambda$ for the beta-Poisson distribution to be well-defined. Based on this restriction and the fact that the model is valid when $N = \infty$, we define an inverse contact parameter $v = N^{-1}$ so that we can fit $v$ over the interval $[0, 1/\hat{\lambda})$. MLEs of $\Phi$ and $v$ are calculated using the BFGS algorithm. We report symmetric 95% confidence intervals for each parameter in each model as well as the

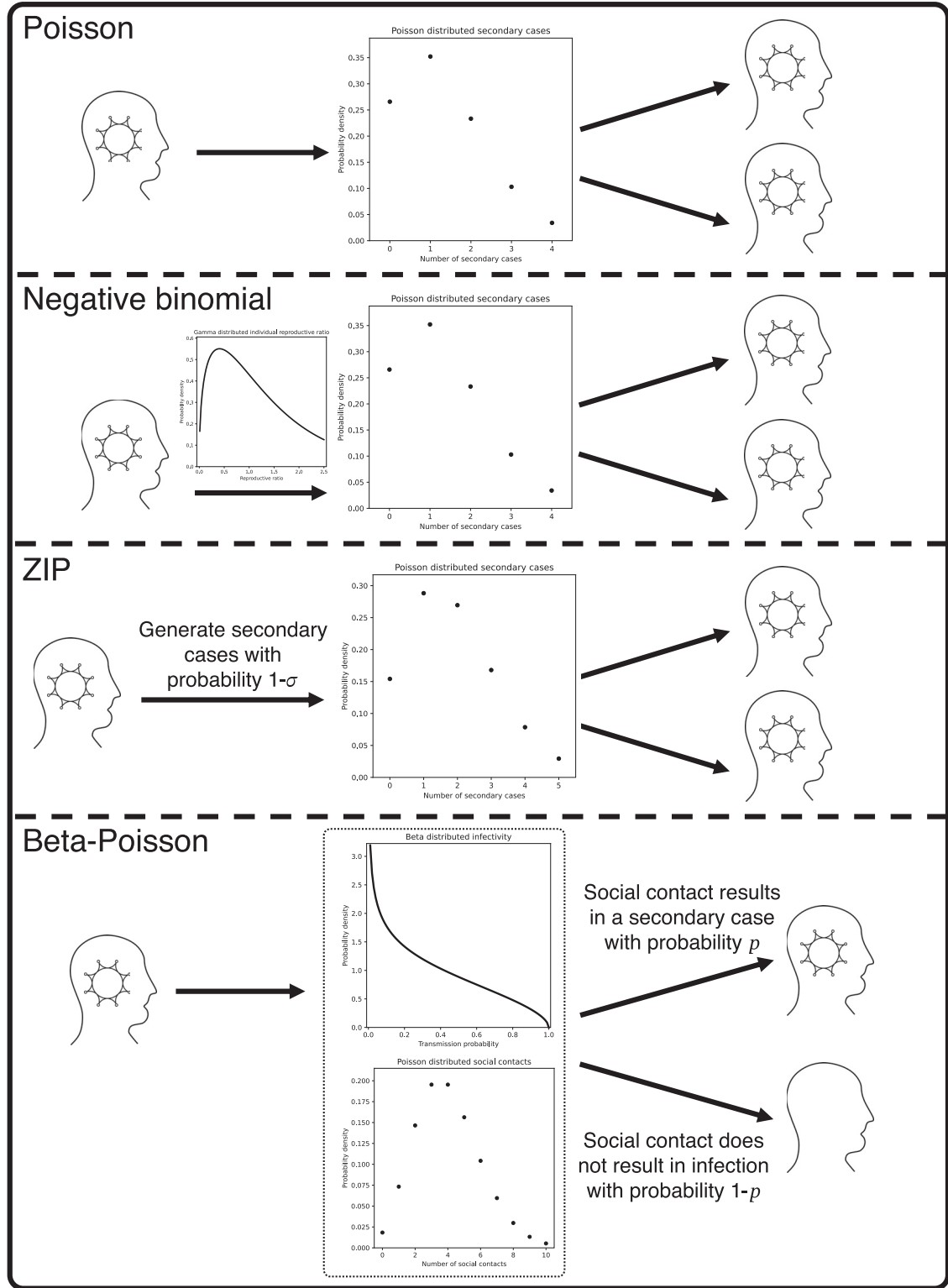

**Fig 1. Mechanistic comparison of the different offspring distribution models.** Under a Poisson model, each infectious individual generates a number of secondary cases, drawn from a Poisson distribution which is the same for all infectious individuals. Under a negative binomial model, each infectious individual is assigned an innate infectivity, drawn from a gamma distribution, which defines a Poisson distribution from which they draw their secondary cases. Under the zero-inflated Poisson model, each infectious individual either generates no secondary cases with probability $\sigma$, or else with probability $1 - \sigma$ generates a Poisson-distributed

number of secondary cases, with the same Poisson parameter for all infectious individuals. Finally, under the beta-Poisson distribution each infectious individual is assigned their own infection probability $p$ from a beta distribution, makes a Poisson-distributed number of contacts, and then infects each of these contacts with probability $p$.

probability of a case generating zero cases, the overdispersion, degree of superspreading by performing 10, 000 bootstrap samples.

In the absence of detailed transmission chain data, branching process models are often parameterised using the total number of cases in an epidemic or set of epidemics [2, 17]. To this end, in Section 6 of S1 Appendix we derive the probability that an epidemic which dies out in its early branching process phase reaches a given size under the beta-Poisson model. The resulting formula involves a complex set of recursive calculations, meaning it is substantially more computationally intensive to calculate likelihoods given outbreak size data in the beta-Poisson model than in the negative binomial model.

## Secondary case data

We fit each candidate model to eight sets of reconstructed secondary case data. Each of the datasets lists the number of cases in an outbreak which are inferred to have generated a given number of secondary cases. Pneumonic plague, mpox (also known as monkeypox), and norovirus are each represented by single datasets. There are two sets of data from Ebola outbreaks and three from outbreaks of infections caused by novel coronaviruses: two of MERS and one of SARS. These datasets include examples of several different transmission routes: pneumonic plague and the novel coronaviruses are spread through airborne transmission, norovirus mostly transmits through fecal-oral contact, and both Ebola and mpox are able to spread both through airborne transmission and through contact with bodily fluids (which itself includes fecal-oral contact).

The plague data is taken from a 2004 paper on pneumonic plague by Gani and Leach and combines data from several outbreaks [34]. This mixture of sources is clearly problematic since it can obscure differences between outbreaks caused by antigenic shift or simply socio-economic disparities between the underlying populations (the data stretches across three continents and ninety years). Gani and Leach themselves find that a geometric distribution gives a good fit to the data, suggesting comparatively homogeneous spreading behaviour despite the range of sources. The mpox data is taken from a 1987 paper by Jezek *et al.* which reported on a mpox surveillance programme in what is now the Democratic Republic of Congo [16]. The

**Table 1. Notation and interpretation of parameters of each model considered in our analysis.**

| Model | Parameter | Range | Interpretation |
|---|---|---|---|
| Poisson | $\lambda$ | $[0, \infty)$ | Basic reproductive ratio |
| Geometric | $\lambda$ | $[0, \infty)$ | Basic reproductive ratio |
| Negative binomial | $\lambda$ | $[0, \infty)$ | Basic reproductive ratio |
| | $\theta$ | $[0, \infty)$ | Degree of overdispersion (variance is $\lambda(1 + \theta)$) |
| ZIP | $\tilde{\lambda}$ | $[0, \infty)$ | Reproductive ratio of cases with "natural" infectivity |
| | $\sigma$ | $[0, 1)$ | Probability a case is prevented from reproducing |
| Beta-Poisson | $\lambda$ | $[0, \infty)$ | Basic reproductive ratio |
| | $\Phi$ | $[0, \infty)$ | Measure of overdispersion ($\theta = \Phi^{-1}$ in negative binomial limit) |
| | $N$ | $[\lambda, \infty]$ | Social contact rate ($N = \infty$ case recovers negative binomial) |
| | $\nu = N^{-1}$ | $[0, 1/\lambda]$ | Inverse social contact rate ($\nu = 0$ case recovers negative binomial) |

**Table 2. Frequency of secondary case numbers by dataset.** Underlined entries denote the superspreading boundary in each dataset.

| Secondary cases | Plague | Mpox | Ebola, Nigeria 2014 | Ebola, Guinea 2014 | SARS, Singapore 2003 | MERS, South Korea 2015 | MERS, Saudi Arabia 2015 | Norovirus, Netherlands 2012 |
|---|---|---|---|---|---|---|---|---|
| 0 | 16 | 163 | 15 | 109 | 162 | 146 | 13 | 22 |
| 1 | 10 | 32 | 2 | 16 | 19 | 10 | 5 | 13 |
| 2 | 7 | <u>10</u> | 1 | 9 | 8 | 4 | 4 | 6 |
| 3 | 2 | 2 | 1 | 5 | <u>7</u> | 1 | 1 | 3 |
| 4 | 3 | 0 | <u>0</u> | <u>5</u> | 0 | <u>0</u> | <u>0</u> | <u>1</u> |
| 5 | <u>1</u> | 1 | 0 | 2 | 0 | 1 | 0 | 1 |
| 6 | 1 | 0 | 0 | 0 | 0 | 1 | 0 | 0 |
| 7 | 0 | 0 | 0 | 0 | 1 | 0 | 1 | 0 |
| 8 | 0 | 0 | 0 | 1 | 0 | 0 | 0 | 0 |
| 9 | 0 | 0 | 0 | 3 | 0 | 0 | 0 | 0 |
| 12 | 0 | 0 | 1 | 0 | 1 | 0 | 0 | 0 |
| 14 | 0 | 0 | 0 | 1 | 0 | 0 | 0 | 0 |
| 17 | 0 | 0 | 0 | 1 | 0 | 0 | 0 | 0 |
| 21 | 0 | 0 | 0 | 0 | 1 | 0 | 0 | 0 |
| 23 | 0 | 0 | 0 | 0 | 1 | 1 | 0 | 0 |
| 38 | 0 | 0 | 0 | 0 | 0 | 1 | 0 | 0 |
| 40 | 0 | 0 | 0 | 0 | 1 | 0 | 0 | 0 |
| 81 | 0 | 0 | 0 | 0 | 0 | 1 | 0 | 0 |
| Total | 53 | 63 | 19 | 145 | 159 | 174 | 23 | 43 |

first Ebola dataset is from a paper by Fasina et al. which constructs the transmission tree of a local outbreak in Nigeria during 2014 [35]. This outbreak was initiated by a single hospitalised patient to whom twelve subsequent cases were traced, suggesting a possible superspreading event. The other Ebola dataset is from a paper by Faye et al. which inferred transmission chains from line list data from the 2014 Ebola outbreak [36]. The line list data included all of the 193 confirmed probable and confirmed cases of Ebola in Guinea up to the time of the study, with 79% of these cases being assigned to transmission chains. The data is thus incomplete, but still provides us with a large sample of transmission behaviour in the epidemic. The SARS dataset is based on a transmission tree constructed by the Centre for Disease Control based on data from a SARS outbreak in Singapore, and contains clear evidence of superspreading behaviour [37]. The first of the two MERS-CoV datasets is from an outbreak of MERS which took place across three hospitals in South Korea in 2015, with each within-hospital outbreak initiated by the same index case who was moved between hospitals [38]. The data is extremely overdispersed, with most cases producing no subsequent cases but the index case producing over 80 subsequent cases. The other MERS-CoV dataset is from an outbreak in Saudi Arabia and is significantly less overdispersed [22]. The norovirus dataset was derived from a transmission tree constructed in a paper by Heijne et al. based on an outbreak in a psychiatric hospital in the Netherlands [39]. This dataset has a sub-geometric level of overdispersion.

The eight datasets are listed in Table 2; secondary case numbers which did not appear in any of the datasets are not listed. In each column the "superspreading boundary" is underlined; this is the first element to appear after the 99th percentile of the Poisson distribution with mean matched to the dataset, so that that entry (if nonzero) and any below it correspond to the superspreading events. All of the datasets contain at least one superspreading event by this definition.

Code for replicating our analysis is available at github.com/JBHilton/beta-poisson-epidemics.

## Results

The MLEs of the beta-Poisson model parameters fitted to each dataset are listed in Table 3 and plotted in Fig 2. The difference in scale of the inverse contact parameter MLE $\hat{v}$ between datasets necessitates a log scale in Fig 2c), meaning that in cases where $\hat{v} = 0$ the MLE and lower 95% confidence bound is not plotted, and in the SARS data both confidence bounds are at zero and so neither the MLE nor the 95% confidence interval are plotted. MLEs and confidence intervals for the negative binomial and ZIP models are provided for reference in Section 9 of S1 Appendix. The AIC achieved by each model fitted to each dataset is listed in Table 4, and the likelihood ratio test statistics obtained by comparing the maximum likelihood attained by the beta-Poisson to those obtained by each of the other models are listed in Table 5. In Section 10 of S1 Appendix we perform a sensitivity analysis on the beta-Poisson model by exploring the likelihood surface of the model parameters around each MLE. As a generalization of the other four models, any likelihood obtained by these models is attainable by the beta-Poisson through a suitable choice of parameters and so for each dataset the highest likelihood across all the models will be returned by the beta-Poisson model with maximum likelihood parameters. Despite this, Table 4 reveals that the beta-Poisson is never the optimal choice in terms of AIC, with the improvement in likelihood never substantial enough to justify the addition of an extra parameter relative to the negative binomial distribution. For secondary case distributions with low levels of overdispersion (plague, mpox, norovirus, MERS in Saudi Arabia, see Fig 3) the geometric distribution attains the smallest AIC, while the high likelihood ratios between the geometric and beta-Poisson models do not support the rejection of the geometric model. In the other four cases (the two Ebola datasets, SARS, and MERS in South Korea), the negative binomial model is optimal in terms of AIC, and the likelihood ratio between the beta-Poisson and negative binomial models is close to 1. In particular, both SARS and the South Korean MERS outbreak saw one or more dramatic superspreading events where a single individual is responsible for a substantial proportion of the observed cases, reflected by very narrow confidence intervals around zero for the beta-Poisson model's inverse contact parameter $v$.

The lower portions of the empirical secondary case distributions and model distributions with fitted MLE parameters are plotted in Fig 4. For the mpox data, Singapore SARS data, and both sets of MERS data the maximum likelihood distribution is identical for the negative

**Table 3. Maximum likelihood estimates of beta-Poisson model parameters by dataset, 95% confidence intervals in parentheses.** Values of 0.00 in the confidence intervals for Φ represent small non-zero values rounded for display purposes; values of 0 in the MLE or confidence intervals for $v$ represent "true" zeros corresponding to fits where the MLE is at the negative binomial limit.

| Dataset | λ | Φ | $v$ |
|---|---|---|---|
| Plague | 1.32 (0.88, 1.82) | 0.58 (0.00, 7.88) | 0.25 (0, 0.51) |
| Mpox | 0.3 (0.22, 0.4) | 1.93 (0.00, 9.31) | 0 (0, 1.68) |
| Ebola, Nigeria 2014 | 0.95 (0.15, 2.3) | 0.12 (0.00, 10.22) | 0.05 (0, 6.67) |
| Ebola, Guinea 2014 | 0.95 (0.61, 1.36) | 0.18 (0.1, 0.34) | 0.01 (0, 0.14) |
| SARS, Singapore 2003 | 0.79 (0.36, 1.38) | 0.12 (0.05, 0.42) | 0 (0, 0) |
| MERS, South Korea 2015 | 1.05 (0.2, 2.31) | 0.04 (0.01, 0.4) | 0 (0, 0.02) |
| MERS, Saudi Arabia 2015 | 0.96 (0.42, 1.62) | 0.75 (0.00, 12.41) | 0 (0, 1.64) |
| Norovirus, Netherlands 2012 | 0.93 (0.61, 1.28) | 1.07 (0.00, 16.52) | 0.31 (0, 1.18) |

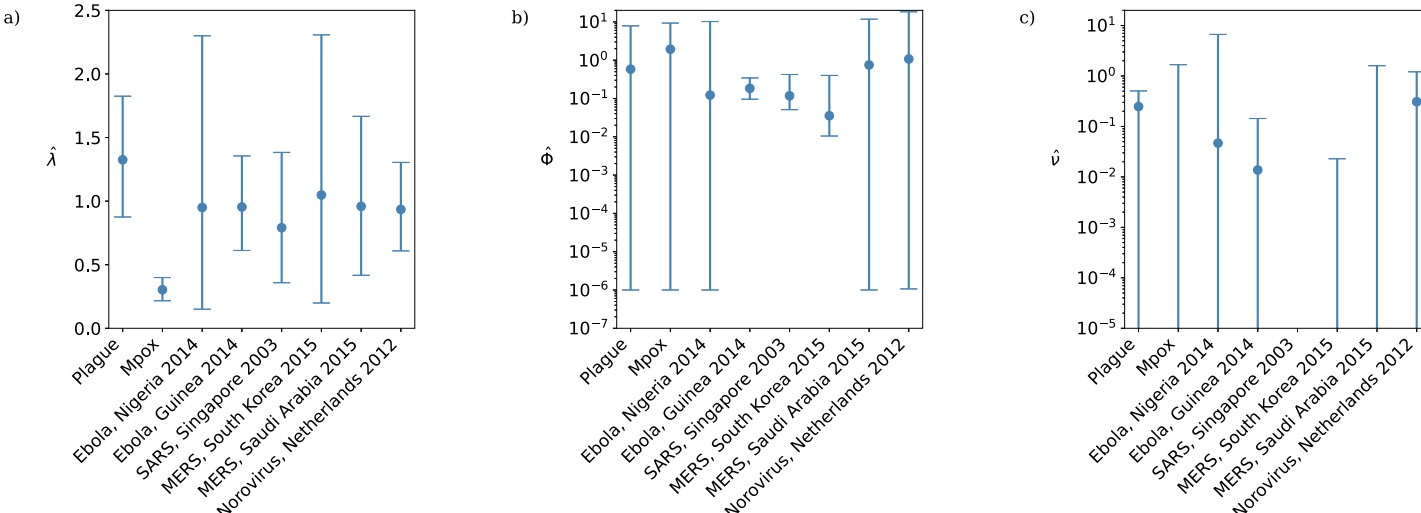

**Fig 2.** Maximum likelihood estimates of the beta-Poisson model parameters by dataset: a) basic reproductive ratio λ; b) overdispersion parameter Φ; c) inverse contact parameter $\nu$. Black lines are 95% confidence intervals. In plot c) MLEs and lower confidence bounds of 0 are not shown; these results should be compared with Table 4 which identifies these cases.

binomial and beta-Poisson distributions, and so only the fitted negative binomial is shown. In those cases where the beta-Poisson does not fit to its negative binomial limit, the fitted beta-Poisson distribution does not appear to differ dramatically from the fitted negative binomial in its description of the transmission potential of low-progeny cases. Fig 3 suggests that where it does not fit to its negative binomial limit (i.e. $\hat{\nu} \neq 0$) the beta-Poisson performs similarly to the negative binomial in its ability to capture the level of overdispersion in the data, although with slightly narrower 95% confidence intervals. The overdispersion in the secondary case datasets is driven both by superspreading events and by an abundance of cases producing zero secondary cases. Fig 5 reveals that the beta-Poisson and negative binomial models both predict similar proportions of superspreaders when fitted to each dataset, and that in general these proportions are not necessarily close to those observed in the data. Fig 6 demonstrates that both of these models fit closely to the observed proportion of zero-progeny cases, as does the ZIP distribution. Note that for the mpox data, Singapore SARS data, and both sets of MERS data the maximum likelihood distribution is identical for the negative binomial and beta-Poisson distributions and so the values plotted for those two distributions in Figs 3, 5 and 6 are identical.

**Table 4. Akaike information criterion at MLE by model and dataset.** The underlined entry of each row is the minimal value of AIC attained for that dataset. Values in brackets in column names give the number of parameters which are inferred for each model.

| Dataset | Poisson(1) | Geometric(1) | Neg. bin.(2) | ZIP(2) | Beta-Poisson(3) |
|---|---|---|---|---|---|
| Plague | 136.84 | <u>129.1</u> | 130.62 | 131.89 | 132.24 |
| Mpox | 309.1 | <u>295.9</u> | 296.63 | 298.02 | 298.63 |
| Ebola, Nigeria 2014 | 86.89 | 56.04 | <u>47.45</u> | 58.47 | 49.41 |
| Ebola, Guinea 2014 | 602.41 | 413.56 | <u>358.39</u> | 416.11 | 360.37 |
| SARS, Singapore 2003 | 902.35 | 496.15 | <u>357.47</u> | 579.32 | 359.47 |
| MERS, South Korea 2015 | 1230.77 | 473.15 | <u>224.66</u> | 618.44 | 226.66 |
| MERS, Saudi Arabia 2015 | 76.14 | <u>67.13</u> | 68.89 | 71.84 | 70.89 |
| Norovirus, Netherlands 2012 | 128.8 | <u>125.28</u> | 126.57 | 127.25 | 128.46 |

**Table 5. Likelihood ratio test statistics obtained by comparing beta-Poisson to other candidate models under each dataset.** Values in brackets in column names give the difference in number of parameters between each model and the beta-Poisson distribution.

| Dataset | Poisson(2) | Geometric(2) | Neg. bin.(1) | ZIP(1) |
|---|---|---|---|---|
| Plague | 0.01 | 0.65 | 0.54 | 0.2 |
| Mpox | 0.0 | 0.53 | 1.0 | 0.24 |
| Ebola, Nigeria 2014 | 0.0 | 0.0 | 0.83 | 0.0 |
| Ebola, Guinea 2014 | 0.0 | 0.0 | 0.87 | 0.0 |
| SARS, Singapore 2003 | 0.0 | 0.0 | 1.0 | 0.0 |
| MERS, South Korea 2015 | 0.0 | 0.0 | 1.0 | 0.0 |
| MERS, Saudi Arabia 2015 | 0.01 | 0.89 | 1.0 | 0.09 |
| Norovirus, Netherlands 2012 | 0.11 | 0.66 | 0.74 | 0.38 |

## Discussion

In this study we have introduced a beta-Poisson mixture model for infectious disease transmission with two desirable properties: it is a mathematical generalization of both the negative binomial and ZIP models, and its interpretation as a combination of distinct contact and transmission steps means that it can be used to model changes to contact behaviour while preserving individual-level variation in infectivity. This interpretation offers a key advantage over the negative binomial distribution, where it can be unclear how the impact of a control measure should be reflected in the parameters of the underlying gamma distribution. As a generalization of the negative binomial the beta-Poisson distribution should offer an improvement in its ability to capture the transmission patterns seen in real-world outbreaks, although our results suggest that this improvement is slight. In particular, when we performed model selection using AIC, the beta-Poisson distribution was not found to be the optimal model for any of the transmission chain datasets we analysed. In addition to this, for the datasets with lower levels of overdispersion the estimates of the parameter $\Phi$ in the beta-Poisson model had very wide confidence intervals, suggesting that the transmission chain data was not informative enough to parameterize this model. Thus, while the beta-Poisson model still has the potential to be useful in modelling non-pharmaceutical interventions, which reduce average contact rates, the negative binomial is likely to be preferable as a descriptive model of observed outbreaks and as a predictive model of uncontrolled transmission. One of the conceptual limitations of the beta-Poisson model is that while individual-level infectiousness is strongly heterogeneous, each case's social contact behaviour follows the same Poisson distribution and so individual-level variations in gregariousness are not captured by the model. Although combining a beta distributed infectivity with a non-Poisson contact distribution (for instance, a negative binomial) would capture this additional level of variation, the wide confidence intervals for the parameter $\Phi$ and minimal improvement in likelihood in the fits for the beta-Poisson model suggest that a more complex model in this vein would be difficult to parameterise confidently while also being unlikely to offer a meaningfully better fit to data than existing models. Our findings are consistent with Kremer *et al.*'s analysis of the Poisson-generalized gamma distribution, which found that specific two-parameter subcases were preferable in terms of AIC to their more general three-parameter distribution as a model for COVID-19 contact tracing data [29].

Our assessment of the beta-Poisson model's descriptive power focussed on transmission chain data, but in the absence of this level of detail in the data, branching process models can be parameterised using records of the total size of a set of localized outbreaks [17]. The presence of the explicit contact parameter $N$ in the beta-Poisson model introduces complexities in

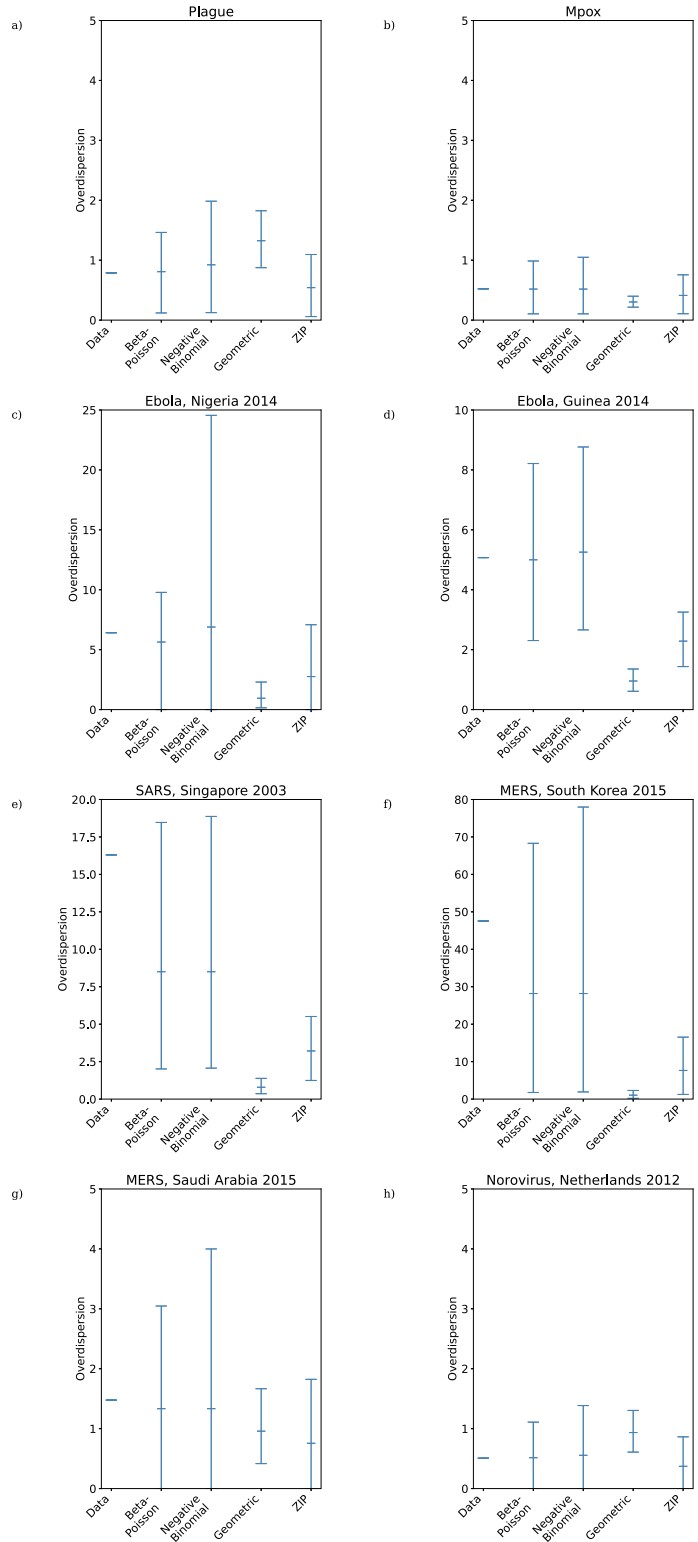

**Fig 3.** Overdisperion of maximum likelihood offspring distributions fitted to reconstructed transmission trees from a) plague; b) Mpox; c) Ebola, Nigeria 2014; d) Ebola, Guinea 2014; e) SARS, Singapore 2003; f) MERS, South Korea 2015; g) MERS, Saudi Arabia 2015; h) Norovirus, Netherlands 2012. Black lines are 95% confidence intervals.

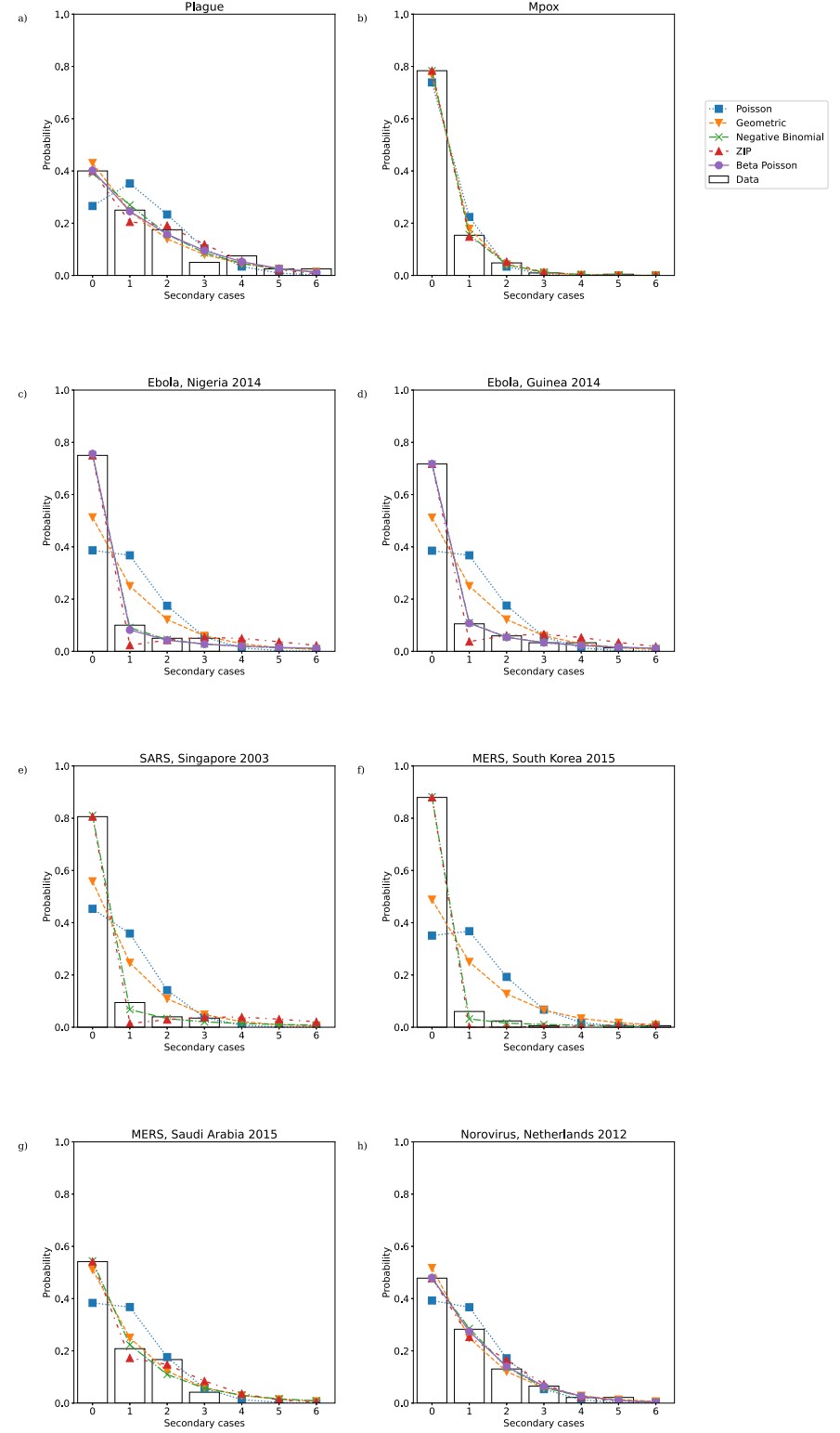

**Fig 4.** Lower portion of maximum likelihood offspring distributions fitted to secondary case data from a) plague; b) Mpox; c) Ebola, Nigeria 2014; d) Ebola, Guinea 2014; e) SARS, Singapore 2003; f) MERS, South Korea 2015; g) MERS, Saudi Arabia 2015; h) Norovirus, Netherlands 2012.

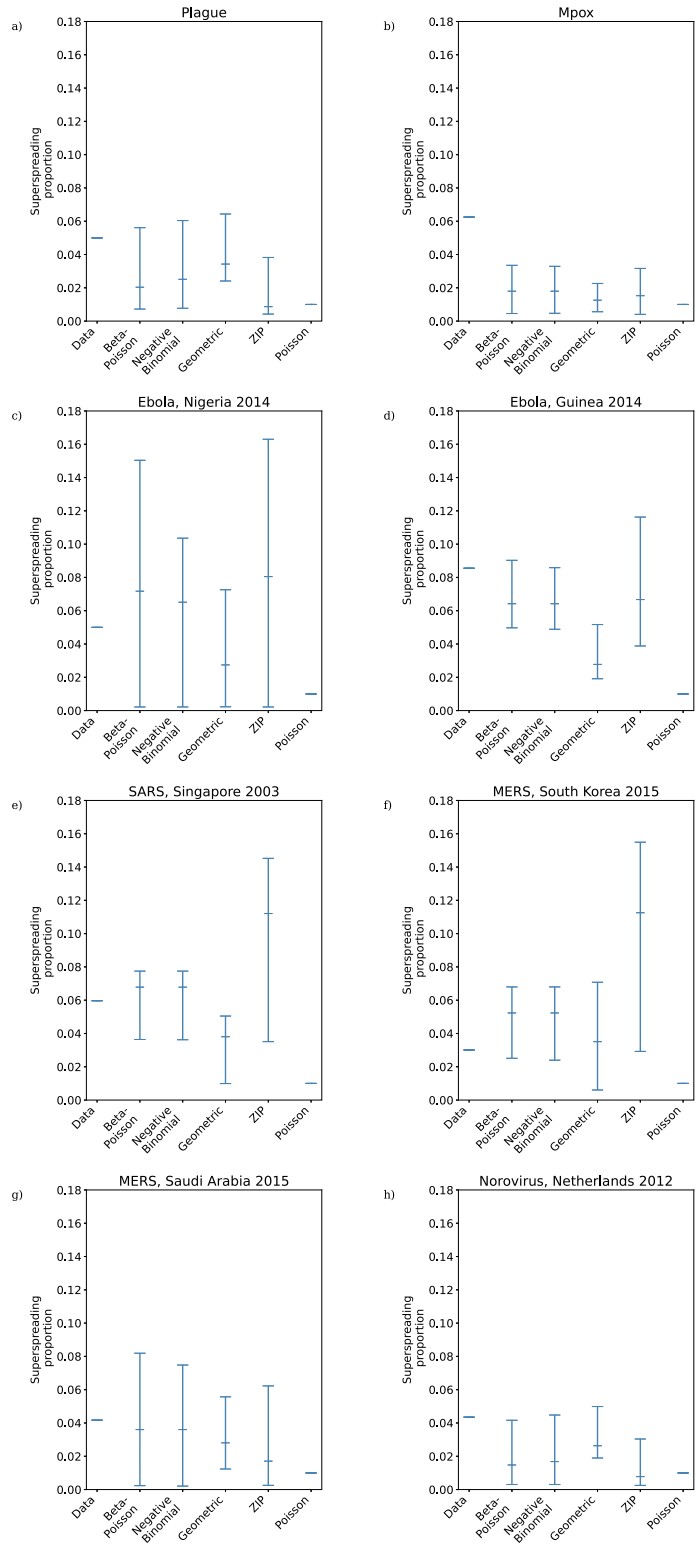

**Fig 5.** Proportion of superspreaders in maximum likelihood offspring distributions fitted to reconstructed transmission trees from a) plague; b) Mpox; c) Ebola, Nigeria 2014; d) Ebola, Guinea 2014; e) SARS, Singapore 2003; f) MERS, South Korea 2015; g) MERS, Saudi Arabia 2015; h) Norovirus, Netherlands 2012. Black lines are 95% confidence intervals.

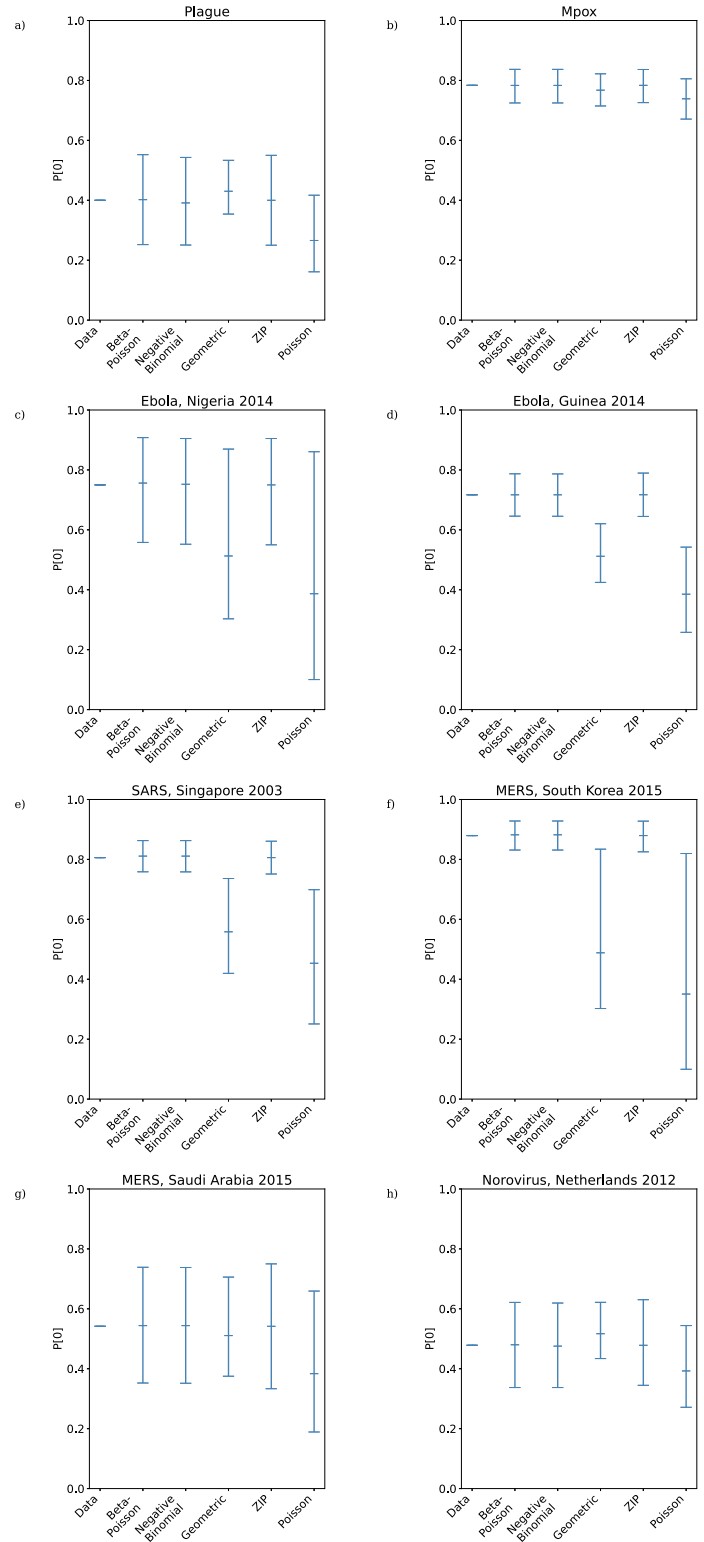

**Fig 6.** Proportion of zeros in maximum likelihood offspring distributions fitted to reconstructed transmission trees from a) plague; b) Mpox; c) Ebola, Nigeria 2014; d) Ebola, Guinea 2014; e) SARS, Singapore 2003; f) MERS, South Korea 2015; g) MERS, Saudi Arabia 2015; h) Norovirus, Netherlands 2012. Black lines are 95% confidence intervals.

dealing with outbreak size data, since there is no a priori reason to think that this *N* will stay fixed across disparate population settings. This means that any dataset which include the sizes of outbreaks in multiple distinct settings will potentially include samples from distributions with different underlying parameters, which is particularly problematic in situations where data is combined from multiple countries and/or over long time intervals. The same concern also applies to the parameters of the beta distribution describing individual-level infectivity, since outbreaks in different segments of the same population will be subject to different variations in infectivity. For instance, children are known to shed greater quantities of influenza virus than adults [40] and contact surveys also suggest that a higher proportion of their contacts are physical than those of adults [30], meaning the infectivity of children will follow a different distribution to that of adults. This is particularly relevant when working with transmission chains inferred from outbreaks in hospitals, where social contact structures are likely to be unrepresentative of those in the general population, particularly along age-structured lines since children and adults are usually assigned to separate wards. Of the example datasets we studied here, the Nigerian Ebola [35], South Korean MERS [38], Saudi Arabian MERS [22], and Dutch norovirus [39] datasets all come from sources which mention hospital-based transmission, meaning our parameterizations may not lead to realistic dynamics outside of a hospital setting. While the presence of an explicit social contact parameter make these limitations obvious in the beta-Poisson model, the same limitations apply to the negative binomial model and indeed to any epidemiological model fitted to real-world data since the parameters of these models will also encode both disease-specific and population-specific factors. These considerations do not rule out the possibility of using models fitted from a specific subpopulation to project the dynamics of an outbreak in a more general population, but they do emphasise the need to critically assess the representativeness of the populations from which outbreak data is taken.

## Supporting information

**S1 Appendix.** Fig A. Log-likelihood curves of the beta-Poisson model parameters by dataset. Table A. Maximum likelihood estimates of negative binomial model parameters by dataset. Table B. Maximum likelihood estimates of zero-inflated Poisson model parameters by dataset. (PDF)

## Acknowledgments

The authors thank Louise Dyson and Jonathan Read for their helpful comments on this work.

## Author Contributions

**Conceptualization:** Joe Hilton, Ian Hall.

**Data curation:** Joe Hilton.

**Formal analysis:** Joe Hilton, Ian Hall.

**Investigation:** Joe Hilton.

**Methodology:** Joe Hilton, Ian Hall.

**Software:** Joe Hilton.

**Supervision:** Ian Hall.

**Visualization:** Joe Hilton.

**Writing – original draft:** Joe Hilton, Ian Hall.

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
