## [Decision Letter · Decision Letter 0]

12 Jul 2023

Dear Professor Hall,

Thank you very much for submitting your manuscript "A beta-Poisson model for infectious disease transmission" for consideration at PLOS Computational Biology.

As with all papers reviewed by the journal, your manuscript was reviewed by members of the editorial board and by several independent reviewers. In light of the reviews (below this email), we would like to invite the resubmission of a significantly-revised version that takes into account the reviewers' comments.

We cannot make any decision about publication until we have seen the revised manuscript and your response to the reviewers' comments. Your revised manuscript is also likely to be sent to reviewers for further evaluation.

Sincerely,

Eric Lofgren, MSPH, PhD

Academic Editor

PLOS Computational Biology

Virginia Pitzer

Section Editor

PLOS Computational Biology

Reviewer's Responses to Questions

**Comments to the Authors:**

Reviewer #1: The authors propose a beta-Poisson approach to characterizing the offspring distribution in an outbreak. I feel the paper touches on potentially interesting and important topic and am fine with the authors reporting negative findings; however, the manuscript is rather poorly written and the motivation, implications, exact procedures and results of the study were not clear enough. For this reason, I would like to encourage the authors to substantially rewrite the manuscript so that the key findings and messages of the paper are clearer (see comments below for possible directions for doing so). I regret to note that it is difficult to conclude that the paper is worth publishing before seeing the improved version of the manuscript. Especially, most part of the current results section seems to be actually explaining methods, and I am not sure if substantial findings remain in results after the authors move the methodological part to methods. Another question is whether the paper provides substantial increment compared with Kremer et al. (https://www.nature.com/articles/s41598-021-93578-x)

Major comments

- While the authors nicely summarize the current practices and backgrounds regarding modelling of overdispersion until L67, the following transition to the beta-Poisson mixture, the topic of this paper, reads rather abrupt and awkward. In particular, the motivation for this study (i.e., why the authors thought the current approaches may be insufficient and were interested in beta-Poisson) should be provided.

- I also did not fully get the objectives of the paper from reading the introduction section. Either the “effectiveness” in L83 or “performance” in L85 was not defined and I do not know what the authors mean here.

- Results: I see this section is not really pure results and the authors explain a lot of background and methods along with the results, which I feel is making it challenging for the readers to take in. Meanwhile, the current method section contains obvious demonstrations that aren’t necessarily crucial in the main text. For example, it’s nice that the authors provide step-by-step explanation of how the beta-Poisson model can be constructed, but a typical reader of PLoS Comp Bio would know what it is just from reading L75–77 or L92–95. I would suggest the current method section is moved to the supplementary materials entirely and the methodological explanations in the current results section are instead moved to the method section. That way, the readers would better understand what the authors did in this paper, rather than each mathematical concept that appears in the methodology. On that note, it would also be helpful to split the results section into multiple subsections.

- As noted above, Kremer et al. has studied a similar (though with a different interest and direction) topic. The authors may not have been aware of this paper as it is not included in the reference list, but I think it would potentially inform how the authors should frame this paper.

Minor comments

- L9–13: I am not sure if this is really the case in real-world outbreaks. There may well be overlooked cases in the earliest phase of a spillover event; contact tracing could also be challenging. Do the authors have any evidence supporting their statement?

- L14: “its mean”: I don’t get what mean is being discussed here

- L88–89: this would be better place at the end of the methods section.

Reviewer #2: This paper introduces the beta-Poisson mixture model as an alternative approach to the negative binomial and zero-inflated Poisson to model the number of secondary cases produced by an infectious individual. The authors offer a comprehensive description of the beta-Poisson model, moment calculations, likelihood calculations, and code to reproduce the results presented in the paper. The authors fit the beta-Poisson distribution to eight different datasets on secondary cases and compared these fits to data also fit to the Poisson, zero-inflated Poisson, geometric, and negative binomial distributions. They found that the negative binomial distribution outperforms the beta-Poisson when comparing AIC values for model fitting.

Overall, the paper is interesting from a statistical standpoint and has the potential to be very useful for many different applications to the study of infectious disease and models of infectious disease, particularly when diseases have highly stochastic transmission dynamics. This is a well thought out paper, and the statistical theory and results from the model fitting are robust. I think the text is the main part of the manuscript that needs to be revised. The authors downplay the usefulness of the beta-Poisson over other distributions to such a degree that it leaves the reader wondering about the point of considering this new model. Below I have comments and suggestions for re-organizing the paper to make the message of the paper clearer as well as other comments to improve the manuscript.

Major Comments:

-One of the overall findings of the paper is that even though the beta-Poisson achieves a closer fit to the data, in most cases the negative binomial outperforms the beta-Poisson in terms of the AIC. This is stated in the abstract and in lines 203-205 in the Results when referring to Table 2. I think in an attempt not to oversell their results, the authors conclude that the beta-Poisson is never the optimal choice. This is true, but the difference in AIC values in Table 2 between the negative binomial and the beta-Poisson never exceeds 2, and Table 4 shows that the likelihood ratios are not significant. It really comes down to the extra parameter in the beta-Poisson. The beta-Poisson also consistently outperforms the Poisson, geometric, and ZIP distributions, and these distributions are commonly used. It might be appropriate to reserve the beta-Poisson for special cases, but I would like to see the authors take a stronger position in favor of the beta-Poisson and give more reasoning and concrete examples of why the beta-Poisson is interesting and useful in the context of intervention modeling. I think the beta-Poisson’s inclusion of a social contact parameter is interesting and important, and the way in which this arises in the calculation of the extinction probability in section 3 of the supplement seems very useful.

-The fact that the beta-Poisson distribution has three parameters, makes this distribution more cumbersome than the negative binomial, and in some cases the overdispersion parameter, phi, is unidentifiable. This is evident in the fitting behavior presented in the supplement. The fact that phi is hard to identify does not seem surprising since this parameter always appears as a product of either lambda or N. This could be why your 95% CIs in Table 3 are wide for phi. This phenomenon should be addressed/discussed in the manuscript. In addition, if you use the formulation of the beta-Poisson as it is written in Equation 1 in the main text and find the MLEs alpha_1, alpha_2, and N for the data, does this provide a better fit than making the substitution for phi? Would it be possible to fit the data and make the substitution for phi after obtaining the MLEs?

-There needs to be a stronger concluding paragraph or section that sums up the take-home message and key findings. The reader is left wanting more at the end of the discussion.

Minor Comments:

-In the third paragraph of the introduction around line 47, it would be very helpful to have specific examples with citations of how these distributions have been incorporated in SIR models.

-Understanding superspreading events is one of the goals of fitting these models to data. It think it would be worth mentioning superspreading events in the first paragraph of the introduction when you are explaining the processes that are important to understanding transmission events.

-Line 97 and Line 199: generalisation should be generalization.

-Line 121-124: Maybe I am wrong, but I think that the beta-Poisson distribution can be valuable in more contexts than just being a model that “fits the data better.” For example, the beta-Poisson might be valuable in simulation-based modeling to generate this type of transmission chain.

-Line 129: Which algorithm did you use in this package to find MLEs? This should be stated in this sentence.

-Figure 1: I really like this figure. It is an excellent depiction of how each of the distributions model transmission. The graphs in each panel need to be defined. In the negative binomial panel, for example, the function with the solid line represents the gamma distribution and the dotted function represents the secondary cases. It would be very helpful if these were labeled (same with the beta-Poisson panel). In the last panel on the beta-Poisson, the individuals without the disease are confusing to the reader because it looks like the disease chain stops. I think it would be clearer if the authors kept only the two individuals where one arrow went to the secondary infected individual with probability p and the other arrow when to the uninfected contact.

-Line 187-188: It would be helpful if you use the term “superspreading boundary” in this sentence so we have a definition for this term when it appears in the caption for Table 1.

-Table 4: It would be helpful to add degrees of freedom for the likelihood ratios in this table so they can be interpreted using Wilk’s theorem.

-I am not sure about the formatting requirements of the journal, but I do think the paper might be clearer if the methods section with a description of the model and the data sets came before the results and the discussion.

-The parameter symbols and their definitions are hard to keep track of. For example, the authors use several terms that describe a measure of overdispersion (phi in line 91 and theta=1/phi in line104), and N and nu=1/N in line 137. It would be helpful to have a table containing all the parameters, their symbols and their definitions to accompany the model description.

-It would also be helpful to have a table of the five distributions that you are comparing that contains the parameters that are present in each model and states how many parameters in each model are being estimated. This would greatly help with the AIC interpretation.

-Line 209: I am confused here about why you would reject the geometric distribution. I would interpret the results in Table 4 to mean that the goodness of fit is indistinguishable between the different models. This should be clarified.

-Line 238: “The beta-Poisson distribution will be definition achieve a closer fit to transmission chain data…..” why is this true?

-Line 240: “but our results demonstrate that these models outperform it equally consistently on grounds of parsimony.” This sentence is very confusion and needs to be revised. What are “these models” and what is “it”?

-Line 241: Would there ever be a case when the topology of transmission chains is NOT simple enough to be summarized by two parameters?

-Line 275: The parameters lambda and theta have taken on different definitions throughout the paper. Define them again here in this sentence.

-In Figure A in Section 8 of the supplement, I do not understand why nu = 0 in the third column. In the figure caption it says that nu is fit to the MLE. This should be clarified.

Reviewer #3: Review of manuscript

"A beta-Poisson model for infectious disease transmission"

(by Joe Hilton & Ian Hall)

For PLOS Computational Biology

The manuscript explores the use of a fairly broad/flexible class of probability distributions as the offspring distribution for generation-based branching process approximation of an SIR type epidemic model in a homogeneously mixing population. There are some theoretical connections showing that this class of distributions contains many families of distributions often considered in this context; but the main focus is on fitting the model to data and thereby exploring the potential value of the beta-Poisson model in comparison to the aforementioned more standard families of probability distributions.

Overall the manuscript explores an interesting and worthwhile topic and the results it presents seem likely to be of interest to the epidemic modelling community. My sense though is that the fairly mathematical/statistical nature of the manuscript is not well-suited to the "results first, methods later" format of this journal. Presenting the results and discussion first begs a lot of questions about the methods that can only really be answered in later in the manuscript, which feels very unsatisfactory to this reader. This is not helped by the lack of signposting for the reader, e.g. short paragraphs at the start of a section outlining the content/purpose of that section, which means that reading the manuscript one is often unsure of what to expect next. In summary: there's clearly some good content here, but the present manuscript does not present a clear story/narrative to guide the reader through that content.

For example re the organisation not fitting very well with the content: the results section starts with several paragraphs about the beta-Poisson distribution, its properties, how it relates to other distributions, details of statistical procedures. There are other things too and it's about two pages into the section (plus a 1-page figure) until one actually reaches any results.

There are also many examples of phrasings which are imprecise and mathematical clams which are not justified (see examples below) which make it hard to follow exactly what the logic of the manuscript is in some places.

Overall I find that the manuscript has interesting content but is not really suitable for PLOS Comp Biol. Perhaps with significant re-writing it could be framed appropriately, but I suspect that it would be best written up for a more explicitly mathematical journal.

***

l.14 "its mean": re-word around this phrase. This seems to refer to the mean of the network, which is not a well-defined concept. I think you're trying to refer to the mean of the offspring distribution of the Branching Process implied by the network structure, but that's probably quite hard to explain this early in the manuscript?

l.25: The word "assigned" here probably needs some explanation.

l27-28: The phrase "outbreak size distribution" might need explanation too. The phrasing perhaps implies final outcome, but looking at "before depleted" on l.19 it might also be restricting attetion to outbreaks which are in some sense small compared to the population size?

l.44 "continuous". Usually the mixing distribution is continuous, but not necessarily.

l.95: I assume that each of these contacts which are infected with probability p are infected independently? The independence (given p) is crucial to spell out so the model is clear to the reader. I suggest making this clear by including (preferably here, but I'm a mathematician so elsewhere might actually be more appropriate) the accompanying definition of a beta-Pois variable Y as (Y|X=x,P=p) \\sim Bin(X,P), where X \\sim Poi(N) and P \\sim Beta(a1,a2); with the note that this is equivalent to Y|P=p \\sim Poi(N \\cdot p), where P \\sim Beta(a1,a2). This (i) explicitly connects two versions of the distribution (which I don't think is done at present), the former being nicer for interpretation as done around l.95 and the latter being a little mathamatically simpler and thus nicer for calculations and (ii) makes clear from the latter version that it is a mixed-Poisson distribution. Because of the representation Y|P=p \\sim Poi(N \\cdot p), I wonder if it might be more informative to call this a scaled-Beta-Poisson distribution; since your version uses a scaled Beta random variable (N \\dot P) instead of a Beta random variable (just P) as the mixing distribution for the Poisson variable, i.e. as the distribution for the mean of the conditional Poisson variable.

l.124: What is meant by "their fitting behaviour"?

l.126: What is a confidence interval for a model fit? Do you mean "CIs for these parameters"? Also I do not undestand what "a grid calculation" means; please either explain or give a reference.

l.129: Presumably the CIs are symmetric in the sense of having equal probability in each tail?

l.137: Is it possible to interpret the parameter nu? That would be nice for the reader, even if it's only a bit vague / hand-wavy.

l.190: I think this sentence would be more precise/accurate if "record the number of" is changed to "make up the". It also hinders understanding that the completely different phrasing "superspreading boundary" is used for what seems to be the same thing in the caption for Table 1.

Table 1 & Table 2 highlight particular entries with different methods (bold v underlining). Suggest consistency. It might also be worth mentioning in the discussion that actually several rows of Table 2 have multiple entries quite close to the minimum entry, so which is the "best" model is not clear-cut for some datasets.

l.201-202: I don't know what to understand from the phrasing "if another distribution appeared to offer a higher maximum likelihood". Surely if the beta-Poiss is more general then this is impossible?

l.213: I don't know what it means for a model to fit "decisively".

l.126 "and in these cases": suggest a new sentence here, the sentence is quite long.

Table 3: What do the entries 0.0 mean? Is this just rounding for display? Need to make this clear (or explain/remind why/how the MLE / CI endpoint can be exactly zero).

l.220: This type of explanatory sentence is missing for many other Figures/Tables.

l.227/232/235: Why present a figure and a table with the same information in? Should choose one. I slightly prefer the figures, subject to the comment below about Figure 2.

l.227-228: Not clear to me what "a distinct distribution" means here. Maybe one of the special cases mentioned before? If so, this is not really distinct as the beta-Pois family includes these special cases.

l.238: I don't understand the use of "consistently" here. Surely "always" is correct since beta-Poi is more general than those more common distributions?

l.243: unidentifiability seems to come out of nowhere for me; not mentioned in the results section I don't think.

l.246: "but clearly not infinite" is a phrasing I don't understand: a real-valued parameter cannot be infinite.

l.251: Not clear to me what "Poisson behaviour" means here.

l.261: Does "outbreak size" here refer just to the size of small outbreaks?

l.283: "is" and "implications" plural.singular mismatch.

l.305: "is is" should I think be "it is".

l.308: "contact survey also suggests" needs rewording, perhaps "contact surveys also suggest" is meant?

l.339: To be precise I think this should be "their total number of contacts during this time follows a Poission distribution with mean N".

l.341-342: Independence for different contacts key to mention here.

Display (1): Suggest in the first line leaving e^{-pN} next to (pN)^x / Gamma(x+1) so that the structure P(x|p) P(p) is more obvious. Also this equation needs to have possible values of x indicated.

Display (3): This follows with much easier calculations using the tower property. Depending on formulation used, either of these does that:

P \\sim Beta(a1,a2), Y \\sim Poi(N), P & Y independent; X|Y,P \\sim Bin(Y,P). So E[X] = E[ E[X|Y,P] ] = E[ YP ] = E[Y] E[P] = N a1/(a1+a2). (the second-last inequality being true because P & Y are independent).

P \\sim Beta(a1,a2), Y|P \\sim Poi(N \\dot P). So E[Y] = E[ E[Y|P] ] = E[ N \\dot P ] = N E[P] = N a1/(a1+a2).

(If the equivalence of these formulations has been made explicit then the latter one is probably preferable as it is simpler.)

Display (6): This will follow much more simply using the Law of Total Variance.

l.390: "transmission distribution" phrase not used before I don't think; why introduce it here?

l.393: Is it made explicit what (x_1, \\dots, x_K) means here?

Display (5): Please offer some explanation re where this comes from.

l.395: If the likelihood is not differentiable w.r.t. some of its parameters (it would be good to give a reference for this), then does this not raise lots of questions about the existence and/or uniqueness of MLEs?

l.397: The comment "the branching process structure of our model means .." needs more explanation (or maybe a reference to something similar) to make it clear to the reader what the logic is here.

l.399: What sort of numerical methods can be employed - the non-differentiability mentioned above will presumably impact this?

Figure 2: Why is a bar chart used here? You are not plotting a quantity of something which a bar chart naturally lends itself to. To me a point/symbol for the MLEs, with error bars as they are, would be amuch more natural way to visualise this information. Also I don't think the phrase "inverse contact parameter" is used elsewhere in the manuscript.

Fig 3: Suggest using bigger symbols and thicker lines so the plots are easier to read.

Figs 4-6: As above, though a horizontal line could be used for the data where appropriate, so it is represented in a different way to the MLEs/CIs.

In Supp Mat:

Displays (19) and (20) have variables i on the left hand side and k on the right; these should be the same presumably?

p.5 l.1: The matrix \\bold{D} = (D_{i,j}) should have a range of values specified for i & j.

p.5 l.6: It is not specified which elements of \\bold{D} that \\bold{D}^{m,z} contains.

p.5 just after display (26): "This formula describes a Poisson process" is not correct, and I don't see what it might be meant to mean. A Poisson process is a random process, not a formula which can be evaluated to give a real number.

p.6 last sentence of Sec 6: Re normal approximation for major outbreaks for populations of size 100+, this depends a lot of what tau = Z_\\infty is. Broadly speaking, if tau is close to zero or one then the population size will have to be larger for the normal approximation to be reasonable; but even then how resonabe it is will depend on the particular parameters of the model.

**Have the authors made all data and (if applicable) computational code underlying the findings in their manuscript fully available?**

Reviewer #1: Yes

Reviewer #2: Yes

Reviewer #3: Yes

PLOS authors have the option to publish the peer review history of their article (what does this mean?). If published, this will include your full peer review and any attached files.

Reviewer #1: No

Reviewer #2: No

Reviewer #3: **Yes: **David Sirl
---

## [Decision Letter · Decision Letter 1]

12 Dec 2023

Dear Professor Hall,

Thank you very much for submitting your manuscript "A beta-Poisson model for infectious disease transmission" for consideration at PLOS Computational Biology. As with all papers reviewed by the journal, your manuscript was reviewed by members of the editorial board and by several independent reviewers. The reviewers appreciated the attention to an important topic. Based on the reviews, we are likely to accept this manuscript for publication, providing that you modify the manuscript according to the review recommendations.

The decision of a minor revision is mostly intended to allow you to address the comments by Reviewer 3 and to evaluate whether or not you believe they will improve the manuscript as a whole. Once you have considered these suggestions, we should be able to accept it without further review.

Sincerely,

Eric Lofgren, MSPH, PhD

Academic Editor

PLOS Computational Biology

Virginia Pitzer

Section Editor

PLOS Computational Biology

The decision of a minor revision is mostly intended to allow you to address the comments by Reviewer 3 and to evaluate whether or not you believe they will improve the manuscript as a whole.

Reviewer's Responses to Questions

**Comments to the Authors:**

Reviewer #1: My comments have been addressed

Reviewer #2: I find that the authors have made substantial revisions, and the manuscript has greatly improved. The authors have sufficiently addressed the major concerns that I outlined in the first review and the concerns of the other two reviewers. I only have two minor editorial comments below that the authors need to address.

L 125 Typo here: “Each case draws an individual-level transmission probability p is drawn from from a”. “draws” and “drawn” appear in the same sentence and double “from”.

L319 Comma after interventions

Reviewer #3: Overall my view is that the authors seem to have addressed my (and the other referees'!) concerns quite well. I recommend acceptance subject to addressing the following relatively minor points. There are a number of places in the manuscript where more careful wording would aid the reader and a few points where the changes introduced have resulted in ungrammatical sentences and slight errors in mathematical detail. In this vein, the comments below should also be interpreted also as a guide to the kind of careful proofreading that could significantly improve the manuscript's readability.

Main manuscript

l125. Check wording that uses both "draws" and "drawn" in the one sentence to refer to one "drawing"; also repeated word "from". Possibly just delete "is drawn from".

l132. Check wording "rather without".

l185. "is given by the sample mean" here seems to be a repeat of the information on l178. Gven the detail presented of all the different cases I suggest deleting that first sentence of the paragraph and just presenting the discussion one model at a time.

l190. "report" might be a better word here than "generate"?

l194. Suggest adding comma after "transmission chain data".

ll205--206. Suggest rewording "x secondary cases, for x running ... to any one individual" to avoid the complicated wording involved in the precision here; perhaps to "various numbers of secondary cases"

ll208--210. First mention is of "three from ... coronaviruses", then "the two coronaviruses". If my understanding is correct it might be a bit clearer to remove the word "two" to reduce some possible confusion?

ll247--248. Suggest rewording to "... the 99th percentile of the Poisson distribution with mean matched to the dataset, so that ..."  for improved readability.

l248. "correspond to" might be slightly clearer wording than "make up the" superspreading events?

Table 2 caption: "in bold" needs correcting.

l255. Suggest using "necessitates" rather than the past tense "necessitated".

l256. Missing LaTeX backslash for the Greek character nu.

Table 3. (i) suggest aligning column entries using the decimal point for better readability. (ii) Re the comments about zeros in the caption, might it be possible/simpler to use 0.00 for rounded and 0 for true throughout the table?

l268. Suggest "... reveals that the beta-Poisson is never ..." instead of "reveals that is never" for clarity.

l269. Suggest rewording "substantial" to "substantial enough", "large enough", "enough" or similar.

Table 5 seems very hard to interpret to me. Maybe include corresponding p-values too?

l300. Check repetition of numeral 5 here.

l328. A possibly tidier way to finish this sentence is something like "... a more complex model in this vein would both be difficult to parameterise confidently and be unlikely to fit data meaningfully better than existing models"?

l331. I think that "who" should be "which", as you're referring to the analysis here, not the people.

l335. I wonder if "... transmission chain data but, in the absence of this level of detail in the data, branching process models ..." might be a little clearer?

l343. I wonder if "or" or "and/or" would be better than "and" at the start of this line?

l359&361. I think "fitted" is preferable to / clearer than "fit" on both these lines.

l366. Suggest deleting "would like to" - if you'd like to thank them then go ahead and do it :)

SuppMat

generally (here and in the main manuscript): check and be consistent, you seem to use the term "appendix/appendices" where the journal uses "supplementary material".

l2 Suggest this paragraph before the section heading "1 Model description" - it's clearly not part of that section!

l11 "During their infectious period, ... total number of contacts ... follows a Poisson distribution". Some more care/detail is required here: depending on what the infectious period distribution is the total number of contacts is generally mixed-Poisson. I think that you mean to say that your model assumes that during the infectious period a Poisson number of contacts are made with mean N.

l32 could write the mean in terms of the new parameters too?

l63 the inequality is maybe more helpfully written as N - \\lambda + 1 \\geq \\epsilon \\geq 1 ?

l92 Do you mean "mixed Poisson process" or mixed Poisson *distribution*?

l122 Suggest saying "probability generating function" here, not just "generating function", since there are many kinds of generating function. Maybe abbreviate to PGF at first use and use the abbreviation afterwards?

l169 I think a little more care is needed in formulating this sentence. The formula (35) gives the limiting variance of the scaled major outbreak size, so the standard deviation you describe in words should have population-size dependence so it can tend to zero.

l181 suggest wording "approximate unconditional probabilities of attaining these sizes".

l213 suggest consistency of words/numerals for "zero and 1".

**Have the authors made all data and (if applicable) computational code underlying the findings in their manuscript fully available?**

Reviewer #1: Yes

Reviewer #2: Yes

Reviewer #3: Yes

PLOS authors have the option to publish the peer review history of their article (what does this mean?). If published, this will include your full peer review and any attached files.

Reviewer #1: No

Reviewer #2: No

Reviewer #3: **Yes: **David Sirl

Figure Files:

Data Requirements:

Reproducibility:

References:

---

## [Editor Report · Decision Letter 2]

23 Jan 2024

Dear Professor Hall,

We are pleased to inform you that your manuscript 'A beta-Poisson model for infectious disease transmission' has been provisionally accepted for publication in PLOS Computational Biology.

Best regards,

Eric Lofgren, MSPH, PhD

Academic Editor

PLOS Computational Biology

Virginia Pitzer

Section Editor

PLOS Computational Biology

---

## [Editor Report · Acceptance letter]

1 Feb 2024

PCOMPBIOL-D-23-00486R2 

A beta-Poisson model for infectious disease transmission

Dear Dr Hall,

I am pleased to inform you that your manuscript has been formally accepted for publication in PLOS Computational Biology. Your manuscript is now with our production department and you will be notified of the publication date in due course.

With kind regards,

Zsofia Freund
